# The Impact of AI Scribes on Streamlining Clinical Documentation: A Systematic Review

**DOI:** 10.3390/healthcare13121447

**Published:** 2025-06-16

**Authors:** Maxime Sasseville, Farzaneh Yousefi, Steven Ouellet, Florian Naye, Théo Stefan, Valérie Carnovale, Frédéric Bergeron, Linda Ling, Bobby Gheorghiu, Simon Hagens, Samuel Gareau-Lajoie, Annie LeBlanc

**Affiliations:** 1Faculté des Sciences Infirmières, Université Laval, Québec, QC G1V 0A6, Canada; farzaneh.yousefi.1@ulaval.ca (F.Y.); steven.ouellet@fsi.ulaval.ca (S.O.); frederic.bergeron@bibl.ulaval.ca (F.B.); annie.leblanc@fmed.ulaval.ca (A.L.); 2VITAM—Centre de Recherche en Santé Durable, Québec, QC G1V 0A6, Canada; theo.stefan.ciussscn@ssss.gouv.qc.ca (T.S.); valerie.carnovale.ciussscn@ssss.gouv.qc.ca (V.C.); 3Faculty of Health Sciences, Université de Sherbrooke, Sherbrooke, QC J1K 2R1, Canada; florian.naye2@usherbrooke.ca; 4Canada Health Infoway, Toronto, ON M5H 1J9, Canada; lling@infoway-inforoute.ca (L.L.); bgheorghiu@infoway-inforoute.ca (B.G.); shagens@infoway-inforoute.ca (S.H.); 5Clinique Médical le Carrefour, Groupe de Médecine de Famille, Laval, QC H7T 2P5, Canada; samuel.lajoie@gmail.com

**Keywords:** AI scribes, clinical well-being, healthcare system efficiency, patient engagement, clinical documentation, systematic review

## Abstract

**Background:** Burnout among clinicians, including physicians, is a growing concern in healthcare. An overwhelming burden of clinical documentation is a significant contributor. While medical scribes have been employed to mitigate this burden, they have limitations such as cost, training needs, and high turnover rates. Artificial intelligence (AI) scribe systems can transcribe, summarize, and even interpret clinical conversations, offering a potential solution for improving clinician well-being. We aimed to evaluate the effectiveness of AI scribes in streamlining clinical documentation, with a focus on clinician experience, healthcare system efficiency, and patient engagement. **Methods:** We conducted a systematic review following Cochrane methods and PRISMA guidelines. Two reviewers conducted the selection process independently. Eligible intervention studies included quantitative and mixed-methods studies evaluating AI scribe systems. We summarized the data narratively. **Results:** Eight studies were included. AI scribes demonstrated positive effects on healthcare provider engagement, with users reporting increased involvement in their workflows. The documentation burden showed signs of improvement, as AI scribes helped alleviate the workload for some participants. Many clinicians have found AI systems to be user-friendly and intuitive, although some have expressed concerns about scribe training and documentation quality. A limited impact on reducing burnout was found, although documentation time improved in some studies. **Conclusions:** Most of the studies reported in this review involved small sample sizes and specific healthcare settings, limiting the generalizability of the findings to other contexts. Accuracy and consistency can vary significantly depending on the specific technology, model training data, and implementation approach. AI scribes show promise in improving documentation efficiency and clinician workflow, although the evidence remains limited and heterogeneous. Broader and real-world evaluations are needed to confirm their effectiveness and inform responsible implementations.

## 1. Introduction

Burnout among clinicians, including physicians, is a significant issue in the healthcare system. The overwhelming burden of clinical documentation is a major contributor to this problem [1,2]. According to the Canadian Medical Association, 60% of physicians report an administrative burden as a direct contributor to declining mental health [3]. Excessive time spent on documentation reduces the time available for patient care [4,5] and leads to increased stress, low job satisfaction, and burnout among healthcare providers [1,2,4,5]. Canadian physicians spend approximately 18.5 million hours each year on unnecessary administrative tasks, equivalent to 55.6 million consultations, highlighting the magnitude of this issue [3]. Emotional exhaustion, frustration, and a reduced sense of accomplishment have far-reaching consequences for physicians’ well-being and patient care [1,2,3].

Medical (or human) scribes, defined as professionals who specialize in the real-time documentation of patient-physician interactions during medical examinations, were initially employed to reduce administrative burden [6,7,8]. While they provide significant relief, their high cost, training requirements, and high turnover rates make them unsustainable in many healthcare contexts [9,10]. This has led to the development of digital scribes that use speech recognition technology to transcribe and summarize clinical encounters. Although they have improved efficiency, digital scribes still require human oversight and are limited in understanding the clinical context, leading to errors in complex cases [11,12]. Building on these limitations, the integration of artificial intelligence (AI) into the form of AI scribes has emerged [13,14]. AI scribes, typically powered by large language models or natural language processing algorithms, are designed to generate or support clinical documentation from audio or textual inputs during medical encounters. AI simulates human cognitive functions such as learning, reasoning, and language comprehension [15]. This enables AI scribe systems to automate tasks, understand medical terminology, and assist healthcare professionals in delivering more efficient and accurate care [16,17].

AI scribes are specific applications of AI that use advanced technologies such as speech recognition, natural language processing (NLP), and machine learning to automate clinical encounter documentation [18]. These systems can transcribe and summarize conversations, enabling direct integration into electronic health records (EHRs) with limited human intervention [11,18,19,20]. Unlike human and digital scribes, AI scribes offer a scalable, potentially cost-effective, and sustainable solution that can enhance speed, accuracy, and integration with clinical workflows [9,11,18,19,20]. By understanding medical terminology and context, AI scribes can enable accurate and efficient documentation, reduce clinician burnout, and improve patient care [11,21]. The rise of AI scribes aligns with rapid advancements in NLP and machine learning, which have enhanced their ability to process clinical language and produce accurate documentation [22,23].

AI scribes have begun to be evaluated in clinical settings, showing promising results and highlighting areas that require further assessment. The Permanente Medical Group (TPMG) recently piloted ambient AI scribe technology within an integrated healthcare system to determine the most effective and safe way to integrate AI scribes into clinician workflows [19]. The findings suggest that AI scribes improve physician engagement, enhance the quality of clinical documents, and improve workflow efficiency [19]. While AI scribes offer great promise in terms of workflow improvement and clinician satisfaction, the human element remains crucial, particularly for overseeing more complex cases and addressing unforeseen system limitations [13,24].

As these solutions continue to be used and implemented in clinical settings owing to their clear advantages, there is an urgent need to synthesize existing scientific knowledge on the impact of AI scribes, their technological capabilities, and their integration into healthcare systems. Understanding these factors will provide crucial insights into optimizing the use of AI scribes in clinical environments and maximizing their benefits for both clinicians and patients. In this systematic review, we aimed to describe the effectiveness of AI scribes (defined as automated tools that assist with clinical documentation) by examining their influence on clinician well-being, documentation quality, healthcare system efficiency, and patient engagement in all clinical settings. This is the first systematic review to focus specifically on the implementation and impact of AI scribes in clinical documentation, addressing a critical gap in the literature as this emerging field gains traction in real-world healthcare settings.

## 2. Methods

### 2.1. Overview

The research question of interest was as follows: What are the impacts of AI scribes on clinicians, healthcare system efficiency, documentation, and patient outcomes? This review focused on evaluating AI tools designed to streamline clinical documentation for healthcare providers across all clinical settings. We followed the Cochrane Handbook for Systematic Reviews of Interventions for methodological guidance [25], conducted our review using a PICOS (Population, Intervention, Comparison, Outcomes, Study type) framework, and reported data in accordance with the PRISMA 2020 statement [26]. The PRISMA checklist is presented in Appendix A.

### 2.2. Eligibility Criteria

We included AI-based interventions such as real-time transcription, automated EHR data entry, natural language processing-based clinical summarization, and tools that transform spoken interactions into organized clinical notes. We considered all sources that assessed the following outcomes: clinician outcomes (e.g., overall experience, engagement, documentation burden), healthcare system efficiency metrics (e.g., wait times, patient throughput, costs), documentation outcomes (e.g., accuracy, relevance, reduction in manual note revisions), and patient outcomes (e.g., safety or quality of care and experience with technology). We included all interventional study designs, including RCTs, quasi-experimental designs, prospective cohorts, pre-post studies, observational studies, and mixed-methods studies. There were no restrictions on the language or publication year. We excluded protocols, ongoing studies, qualitative-only studies, reviews, and studies on non-AI interventions. In accordance with our review protocol registered in PROSPERO (CRD42024619680), the PICO(S) elements [26] are presented in Table 1.

### 2.3. Information Sources and Search Strategy

This search strategy was developed in collaboration with an experienced librarian (F.B.). The following databases were searched: Medline (OVID), Embase (Elsevier), CINAHL, and Web of Science. In addition, we searched the preprint repository Arxiv (https://arxiv.org/) and Google Scholar to ensure that we captured relevant articles published in non-health science journals.

We used the expression “AI scribe” and a list of equivalent synonyms such as “Ambient scribe”, “Clinical scribe”, and “Medical scribe. The sensitivity of the search strategy was tested using three key references provided by our partners at Canada Health Infoway, an independent non-profit organization. The details of the search strings used in each database are provided in Appendix B (Table A1). We used the Covidence online software [27] to manage the workflow of this review and an Excel spreadsheet to manage data extraction.

### 2.4. Selection and Data Collection Process

Two reviewers independently conducted the study selection. The screening criteria were pilot-tested on 10 sources, with no discrepancies arising. Any conflicts in decision-making were resolved through discussion or, if necessary, by consulting a third experienced reviewer. Data extraction was carried out by two reviewers using a developed and pilot-tested form. The extracted information was validated by an experienced reviewer using full-text sources. Discrepancies were resolved through discussion or, when necessary, by consulting a senior third reviewer.

The extracted data included study design (e.g., cohort study, pilot study, comparative study, proof-of-concept study, mixed methods, quantitative descriptive, etc.), population characteristics (e.g., number of participants, providers from various specialties, medical students, physicians, nurses, patients, etc.), intervention specifics (e.g., which AI scribe was used, in which context, and for how long), comparator details (e.g., control group characteristics, if applicable), outcomes (e.g., clinician, healthcare efficiency, documentation, and patient-related), and reported impacts (e.g., positive, mixed, neutral, or negative).

### 2.5. Synthesis Methods and Reporting Quality Assessment

We synthesized the data using structured narrative summaries based on our PICOS elements. For qualitative data, a content analysis approach was applied by grouping the information into themes. Our synthesis focuses on providing knowledge users with insights into the impact, best practices, gaps, and challenges associated with AI scribes. The quality of each included review was independently assessed by pairs of reviewers using the Mixed Methods Appraisal Tool (MMAT) [28,29,30], with all scores verified by a senior reviewer. Representatives from Canada Health Infoway actively participated in all stages of this review.

## 3. Results

### 3.1. Study Selection

After removing 362 duplicates (344 by Covidence and 18 manually), 406 titles and abstracts were screened independently. Following exclusion, 20 references remained for the full-text review. Of these, 12 were excluded, leaving eight references suitable for inclusion in this review [31,32,33,34,35,36,37,38]. A PRISMA 2020 flow diagram [39] of the study inclusion process is shown in Figure 1. All included studies were indexed in at least one of the core bibliographic databases searched, and no study was identified solely through Arxiv or Google Scholar.

### 3.2. Study Characteristics

Most of the included studies were conducted in the United States (6/8, 75%), with one study conducted in the Netherlands and the other in Bangladesh. Most studies (6/8, 75%) were published in 2024, with one published in 2023 and the other published in 2021. The main characteristics of the included studies are presented in Table 2.

### 3.3. Methods and Approaches to Evaluating AI Scribes’ Impacts

We identified a range of methods and approaches for evaluating various aspects of the impact of AI scribes. These included two mixed-method pilot studies [32,35], two usability studies [36,37], a comparative study [34], a simulation of patient encounters [38], an AI system development process followed by a post-test questionnaire [33], and a peer-matched controlled cohort study [31].

### 3.4. Settings of Included Studies

The clinical settings across the eight included studies reflected a broad spectrum of environments, emphasizing both real-world and simulated healthcare contexts. Outpatient and ambulatory care settings were represented in two studies [31,35]. One study examined outpatient clinics within an integrated healthcare system [31], while the other focused on clinical settings at a National Cancer Institute designated as a comprehensive cancer center [35]. Simulated environments have also been presented in two studies [34,37]. One study featured simulated patient-provider encounters across various ambulatory specialties [34]. Another study assessed the impact of a Dutch AI scribe system on clinical documentation efficiency and quality by incorporating medical students in a simulated setting at Leiden University Medical Center [37]. Academic and integrated healthcare systems have been identified in three other studies [32,33,38]. One study explored a diverse group of 16 clinicians within an integrated academic healthcare system [32], while another study was conducted in a medical college hospital [33]. One study extended the scope to multiple departments within a university medical center, including two medical students in their analysis [38]. Finally, a specialized context was identified, focusing on phone call recordings from physicians for emergency department referrals at the Nationwide Children’s Hospital Physician Consult and Transfer Center [36].

### 3.5. Global Characteristics of AI Scribes Used in Interventions

The global characteristics of AI scribes used in the interventions are presented in Table 3.

As shown in Table 3, several interventions have focused on ambient and automated transcription technologies. One study highlighted Nuance DAX, a voice-enabled, AI-powered solution that uses conversational AI to capture and document clinical encounters through ambient listening [31]. Another study described the Dragon Ambient eXperience (DAX) smartphone app, which uses AI components to structure recorded information into visit notes with vendor staff performing initial edits [35]. Other studies explored hybrid models that blend AI with human expertise. For example, one study contrasted two virtual scribe solutions: a fully human-driven Live Virtual Scribe and a hybrid Asynchronous Virtual Scribe, which leveraged machine learning and NLP to draft notes from audio recordings that were later refined by medical scribes [32].

We identified two studies that focused on the development and application of AI scribe systems [33,34]. One study introduced an automated AI scribe and intelligent prescribing system capable of converting patient voice inputs into text for medical notes and e-prescriptions [33]. Using NLP and machine learning, it extracts medical terms and integrates prescription capabilities. Another study explored the use of ChatGPT-4 to generate Subjective, Objective, Assessment, and Plan (SOAP) notes, utilizing a standardized clinical documentation format to organize patient information [34].

Advanced language models and speech-to-text systems have also been used. One study utilized a two-step transcription process with AWS Transcribe and various pre-trained language models (e.g., T5, PEGASUS, and BART) to summarize clinical conversations [36]. Another study detailed “Autoscriber”, a multilingual web-based tool that uses transformer-based models and large language models (e.g., GPT-3.5, GPT-4) for transcription and summarization, featuring a unique self-learning capability [37]. Another study described a patient-centered AI scribe designed to automate medical documentation through automatic speech recognition and NLP [38].

### 3.6. Features of AI Scribe Systems

As described in Table 3, we identified six of the most common features of these AI scribe systems: clinical documentation (recording patient information in EHRs) was reported in six studies, real-time transcription (converting spoken language into text as conversations occur) in five, automated notetaking (automatically generating organized clinical notes from interactions) in six, automated data entry (inputting patient data into EHRs without manual intervention) in four, automated summarization (condensing clinical conversations into key points) in six, and administrative assistance (supporting administrative tasks such as scheduling and follow-up) in two studies (out of eight).

### 3.7. Overview of the Impacts of AI Scribe Systems

We identified the impact of AI scribes on each outcome category (clinician, healthcare system efficiency, and documentation) as well as patient outcomes. The effects are listed in Table 4.

As described in Table 4, we identified the impact of AI scribes on clinician outcomes in four studies, including users’ overall experience, healthcare provider engagement, and clinician documentation burden [31,33,35,37]. We identified the effects on healthcare system efficiency in a single study that measured work productivity among AI scribe users and evaluated the attributed panel sizes for value-based care providers [31]. We identified documentation outcomes in seven studies, including efficiency in documentation time and EHR usage, documentation deficiency rates, similarity of system-generated outputs with manually created notes, accuracy and note quality, editing workload, time efficiency, and reliability compared to traditional methods [31,33,34,35,36,37,38]. In three studies, we identified patient outcomes related to their experience with this technology, patient safety, and the role of patient-centered communication [31,35,38].

### 3.8. Impacts of AI Scribes on Clinician Outcomes

AI scribes have demonstrated positive effects on healthcare provider engagement, with users reporting increased involvement in their workflow [31]. The documentation burden showed signs of improvement, as EHR metadata and surveys suggested that AI scribes helped alleviate the workload for some participants [32]. Overall, the user experience was mixed. Many clinicians have found the system to be user-friendly and intuitive, although some have expressed concerns about scribe training and documentation quality [32,33]. During the one-month intervention, the usability, feasibility, and acceptability of AI scribes were assessed across nine survey responses and eight interviews [35]. The results revealed moderate satisfaction but a limited impact on reducing burnout, although perceptions of documentation time improved [35]. Medical students with experience in clinical practice and documentation appreciated the usefulness of the AI tool but found its summaries too rigid, often requiring refinements to improve clarity and relevance [37].

### 3.9. Impacts of AI Scribes on Healthcare System Efficiency

In a single study investigating these impacts, AI scribes were associated with a statistically significant but modest increase in user productivity [31]. However, the practical impact appeared limited, possibly due to the absence of incentives to exceed established organizational expectations. Furthermore, there was no significant change in the attribute panel size for value-based care providers, suggesting that AI scribes did not directly influence patient volume management [31]. In addition, we identified no findings related to healthcare system efficiency metrics such as wait times, patient throughput, and costs.

### 3.10. Impacts of AI Scribes on Documentation Outcomes

AI scribes have contributed to improved documentation efficiency, reducing documentation time per patient while maintaining high similarity between system-generated and manually created notes [31,33]. However, after-hours EHR work increased for users, and documentation deficiency rates showed mixed results, with a reduction in general deficiencies but an increase in CPT submission deficiencies [31]. ChatGPT-4 struggled with accuracy and consistency, particularly in handling non-objective data, highlighting its limitations in clinical documentation [34]. Errors were observed in specific note sections, although the editing workload improved over time, and delays were reported due to vendor release processes [35]. Evaluations of pre-trained models showed moderate performance in identifying key information, with declining accuracy in zero-shot scenarios [36]. A commercial AI scribe system demonstrated improved time efficiency; however, its automatically generated summaries had lower quality scores and higher word counts, requiring refinement [37]. Individual variability in documentation quality and time spent was noted, with some medical students finding automatic summaries more time-consuming to edit than manual documentation [37]. In another study, a patient-centered AI scribe system outperformed traditional methods in terms of speed and reliability, requiring minimal training to enhance clinical workflow [38].

### 3.11. Impacts of AI Scribes on Patient Outcomes

A study investigating AI scribe patient safety concerns reported no adverse events [31]. No significant impact was found on patient satisfaction scores [31], although some patients expressed ease with smartphone-based recordings [35]. A patient-centered AI scribe system showed promise in improving provider-patient communication by enabling seamless documentation without disrupting their engagement [38].

### 3.12. Factors for Successful Adoption and Implementation of AI Scribes in Clinical Settings

We identified the factors for the successful adoption and implementation of AI scribes in clinical settings. Table 5 presents the results.

We identified six main themes. Training and support needs were reported in five studies [31,32,35,37,38], organizational preparation in four [31,32,34,36], technical considerations and improvements in all eight [31,32,33,34,35,36,37,38], evaluation and workflow integration in four [31,32,35,36], and ethical considerations in four [31,35,36,38]. Finally, items for further research and future directions were proposed in all eight studies [31,32,33,34,35,36,37,38].

The successful implementation of AI scribes into clinical workflows requires careful planning, implementation, and training, as outlined in the themes reported in Table 5. Pilot testing and real-world evaluations are recommended to enhance the applicability and measure the effects on workflow, care quality, and patient experience. Technical improvements should focus on building a robust infrastructure, ensuring seamless EHR integration, standardizing procedures, and enhancing usability to enable AI systems to adapt to dynamic clinical environments [32,33,35,38]. A training support framework emphasizing comprehensive and personalized onboarding is important for user adoption and proficiency [31,32,35,36,37,38]. Continuous evaluation is essential and requires iterative feedback loops, ongoing assessments of effectiveness, and refinements based on user needs [32,35,37,38]. Organizational preparation involves aligning AI adoption with institutional priorities, fostering executive sponsorship, and developing operational plans that emphasize accountability and inclusiveness [31,32,36]. All of these themes provide a preliminary roadmap from scientific literature for integrating AI scribes into healthcare.

### 3.13. Patient Perspectives and Ethical Considerations

Patient comfort and privacy are crucial considerations when implementing AI scribe technology [35,36,38]. While some studies suggest that patients generally accept AI scribes, the authors concluded that further research is needed to better understand their perceptions and concerns. Ethical considerations, including data privacy and security, potential bias in AI algorithms, and the impact on the patient-physician relationship, also need to be addressed [35,36,38].

### 3.14. Quality Assessment of Included Studies

We appraised the methodological quality of the eight included studies using the MMAT [28,29,30], and the assigned scores are presented in Appendix C (Table A2).

This peer-matched controlled cohort study enabled a comparative analysis of outcomes, such as caregiver engagement and productivity, involving 99 providers and 76 matched control providers [31]. This study received a score of 3 (out of 5) on the MMAT, as the measurements for both the outcomes and the intervention were not entirely appropriate. These results may have been biased by other factors, making it difficult to make statistically robust assessments of provider engagement. Additionally, the data were incomplete, as the survey data for the control group were unavailable. Although the productivity comparison showed a statistically significant difference, the effect sizes were small [31].

Another study using simulation-based methodologies focused on the efficiency, training requirements, documentation speed, and reliability of a developed AI scribe incorporating patient-centered communication elements [38]. This quantitative, non-randomized study received a score of four stars (out of five). However, it lacks representation of the target population, as the system was tested with only two medical students who alternated between acting as both the physician and the patient, which may not capture the diversity of provider-patient interactions. For example, the two study participants typed at a faster average speed than the typical typing speed for physicians, meaning that the results may not generalize to providers with average or below-average typing speeds [38].

The development and evaluation of AI scribes were highlighted in a Design Science Research (DSR) approach study that focused on the development process [33]. A qualitative requirement elicitation study, consisting of semi-structured interviews with healthcare professionals to gather user requirements, was followed by a quantitative evaluation study that integrated a post-test questionnaire to assess the system’s performance, ensuring that it met clinicians’ needs and enhanced documentation processes. Both the quantitative and qualitative aspects of this study, which were distinct and not conceptualized for integration, received a score of 3 (out of 5) in the MMAT.

A comparative study provided insight into the accuracy and quality of OpenAI’s ChatGPT-4 [34]. This quantitative descriptive study received a score of 4 stars (out of 5), as the sampling strategy was not entirely appropriate for addressing the research question, particularly regarding the inability to draw conclusions about the correlation between types of cases and associated errors. A substantially larger number of encounters is required to better delineate this relationship [34].

One mixed-methods pilot study aimed to explore clinicians’ documentation burden and overall experience with AI scribes, combining both qualitative and quantitative perspectives [32]. This study received strong evaluations for both the quantitative and qualitative methods separately but scored poorly in the mixed-methods section, as the integration of both methods was not effectively and adequately executed. Another mixed-methods longitudinal pilot study assessed changes in clinician well-being and documentation burden over time [35]. This study received high scores for the qualitative section, particularly for the quality of the semi-structured interviews, but low scores for the quantitative part, especially for the survey responses.

A usability study was performed to assess the performance of AI scribes in supporting clinical documentation; a usability study was performed [36]. However, our quality assessment revealed that the data collected, as identified by a “NO” in the S2 question of the MMAT, did not fully address the research objective. This study evaluated four pre-trained LLM models and concluded that the use of the BART-Large-CNN model in clinical documentation has the potential to reduce documentation burden. Although this study scored five stars (out of five) on the quantitative descriptive scale of the MMAT, it lacks qualitative feedback from clinicians to further assess the perceived value and utility of generated summaries and capture semantic completeness [36]. Another randomized usability study assessed the impact of an AI scribe system on clinical documentation efficiency and quality [37]. Evaluated through the quantitative RCT scale of the MMAT, this study received three stars (out of 5), as it was unclear whether randomization was appropriately performed, and the second question on the comparability of the groups at baseline was judged irrelevant.

## 4. Discussion

### 4.1. Summary of Results

Despite the growing interest in leveraging AI to reduce the overwhelming burden of clinical documentation [11,40], no prior systematic review has specifically addressed this topic. This review aims to assess the impact of AI scribes on clinical documentation, clinician well-being, healthcare system efficiency, and patient engagement. Our findings suggest that while AI scribes may reduce documentation time and improve clinician workflow, evidence of their impact on patient outcomes and systemic efficiency remains limited and inconsistent.

Our review indicated that AI scribes showed positive trends in provider engagement, reducing documentation time and may help alleviate the perception of documentation burden for some clinicians [31,32]. Similarly, other studies have found that AI scribes can reduce documentation burden without affecting patient satisfaction [41], and automating clinical documentation can enhance perceived efficiency through NLP-based solutions [42]. However, these benefits are not universal, as studies have generally reported only limited improvements in overall productivity [31].

We provide a comprehensive overview of the benefits and limitations of AI scribes while identifying key areas for future research. Despite the benefits of decreasing documentation time in some cases [31,37,38], AI scribes exhibit variability in performance and limitations. Accuracy remains a key challenge, particularly in complex clinical scenarios or longer transcripts, which often require substantial human intervention for corrections, thus reducing potential time savings [34,35,37]. Additionally, their impact on clinician productivity metrics, such as workload efficiency and patient panel size, is inconsistent [31]. Concerns about patient comfort, ethical considerations, and data security further underscore the need to address these challenges for broader adoption [31,35,36,38]. Our findings align with those of prior studies that highlight the challenges in the accuracy of AI-generated documentation. For instance, significant technical limitations have been reported in NLP systems [11], and inaccuracies in AI scribes often require human oversight, which limits their impact on workflow efficiency [43].

From an implementation perspective, the description of the successful integration of AI scribes depended on several key requirements. These include comprehensive and individualized training programs [31,32,35], technical advancements, such as seamless EHR integration [32,33], and organizational preparation to align AI implementation with institutional priorities [31,32]. The inconsistent performance of some AI models underscores the need for user-centered design and systems thinking to address technical challenges [36], as well as the importance of ongoing evaluation and iterative feedback for continuous refinement [32]. Consistent with findings from our review [35,36,38], the need to establish standards and patient consent processes to maintain trust in clinical settings was also reported in another study [44]. Ethical concerns, including data privacy and patient consent, have been highlighted elsewhere, emphasizing the potential risks to the patient-physician relationship [45].

### 4.2. Strengths

We collaborated with Canada Health Infoway, an independent non-profit organization, throughout all steps of this review to enhance the transferability of the findings. Our sources, all from 2023 to 2024, provide an up-to-date and comprehensive portrait of the rapidly evolving landscape of AI scribe interventions, allowing us to identify both current trends and emerging developments.

### 4.3. Limitations

#### 4.3.1. Limitations of This Review

Our search strategy was not peer-reviewed by another librarian. However, to facilitate replication, we provide detailed documentation in Appendix B (see Table A1). Data from the included studies were initially extracted by two novice reviewers. Subsequently, an experienced reviewer with advanced expertise in methodology and digital health technologies validated all extracted data. Our search was also conducted using English-language terms; thus, only studies with an English title, abstract, or keywords were screened and included, which may have led to the exclusion of potentially relevant studies published in other languages.

#### 4.3.2. Open Issues and Research Gaps Identified in the Included Literature

The studies included in this review had limited sample sizes and were conducted in specific healthcare settings, limiting the generalizability of their findings. The accuracy and performance of AI scribe systems vary significantly, depending on the underlying technology, model training data, and implementation approach. Most studies have focused on proximal outcomes such as documentation time and perceived efficiency gains. A meta-analysis was not performed due to the significant heterogeneity in study designs, intervention characteristics, outcome measures, and reporting formats. As a result, narrative synthesis was conducted to summarize and interpret findings across studies in a structured manner.

Due to the limited number of empirical evaluations on this subject, methodological variability, inconsistent sample size justification, lack of intervention standardization, and heterogeneity in outcome measures weaken the strength of evidence. When the field is developed further, future reviews should consider stricter inclusion criteria and encourage adherence to established reporting guidelines to enhance comparability and validity. Most of the included studies reported favorable outcomes for AI scribes, which raises the possibility of publication bias. Given the small number of eligible studies, we could not formally assess this bias, which should be considered when interpreting the results.

While this review focused on clinical and implementation outcomes, a systematic comparison of AI scribe architectures (e.g., rule-based NLP vs. deep learning models) was not feasible due to inconsistent reporting and high variability in system descriptions across studies. A dedicated technical review is warranted as the field evolves and more studies with standardized, transparent descriptions of algorithmic frameworks become available.

### 4.4. Future Directions

To advance the integration of AI scribes in healthcare, future research should prioritize improving accuracy, identifying and reducing errors, and optimizing the user experience through adaptive learning and feedback mechanisms. Standardized evaluation frameworks and cost-effectiveness analyses are essential for demonstrating their value across diverse clinical settings. Additionally, addressing ethical concerns such as data privacy and algorithmic bias is crucial for fostering trust and promoting widespread adoption. Pragmatic trials and longitudinal studies could further elucidate the long-term impacts of these tools on clinician well-being, workflow efficiency, and patient outcomes. Future studies should place greater emphasis on capturing patient perspectives, particularly in relation to trust, communication dynamics, and perceived quality of care when AI scribes are used, as these dimensions remain critically underexplored and are essential for informed equitable implementation.

For healthcare administrators, AI scribes represent a promising tool for addressing the growing administrative burden that contributes to clinician burnout. However, implementing AI scribe technology, like any complex healthcare intervention, requires careful planning and execution. A key aspect highlighted by these studies was the preparation of an EHR system for integration with the AI scribe solution. This involved creating new note types, templates, and workflows tailored specifically for notes generated by an AI scribe. Effective clinician training, engagement, and commitment were identified as essential facilitators of adoption. Individualized training, particularly one-on-one champion training in the early stages, has been shown to improve the learning curve of providers. Ongoing support, such as having an on-site support person, is also crucial for troubleshooting the technical or workflow issues that may arise during implementation. The implementation of AI scribes raises important ethical considerations that must be addressed to ensure its safe, equitable, and trustworthy use in clinical settings. Data privacy is a central concern, particularly when systems rely on ambient listening or passive audio capture that may involve sensitive patient information. Transparent data governance, secure storage, and clearly communicated consent processes are essential to maintain patient trust. AI scribes trained on limited or biased datasets also risk perpetuating systemic inequities, potentially leading to inaccuracies in documentation for under-represented populations. Therefore, it is critical to assess and mitigate these biases through inclusive model development and continuous monitoring. AI scribes may be less effective or even inaccessible in certain contexts, such as resource-limited settings, multilingual environments, or when patients have speech impairments, thus raising the risk of digital exclusion. Addressing these factors thoroughly during the implementation process can significantly enhance the effectiveness of AI-scribe solutions in clinical practice. Studies have indicated that the accuracy of AI scribes can vary depending on the specific system and context. Some AI systems, especially those incorporating human reviews, can achieve high levels of accuracy, while others still require further development to improve their reliability.

## 5. Conclusions

AI scribes demonstrated the potential to reduce documentation time and alleviate clinician burden in certain settings, particularly when integrated with tailored workflows and supported by individualized training, although improvements in overall productivity and accuracy have been inconsistent across studies. Documentation outcomes showed the most consistent benefits, including enhanced efficiency and quality in clinical notes, while impacts on patient outcomes and healthcare system efficiency were limited or mixed, highlighting the need for further evaluation through pragmatic, real-world studies. Further research is needed to address concerns related to patients’ perspectives, data privacy, algorithmic bias, job displacement, and evolving dynamics of the healthcare workforce in the context of AI adoption. As thoughtfully integrated AI scribes have become more widespread, future studies could provide valuable data on long-term outcomes, enhancing our understanding of their effects on patient care, cost-effectiveness, and environmental sustainability. Given the high perceived value of AI scribes, their adoption is often accelerated, underscoring the urgent need for better preparation, including robust implementation strategies, stakeholder engagement, and system-level readiness to ensure sustainable and effective integration.

## Figures and Tables

**Figure 1 healthcare-13-01447-f001:**
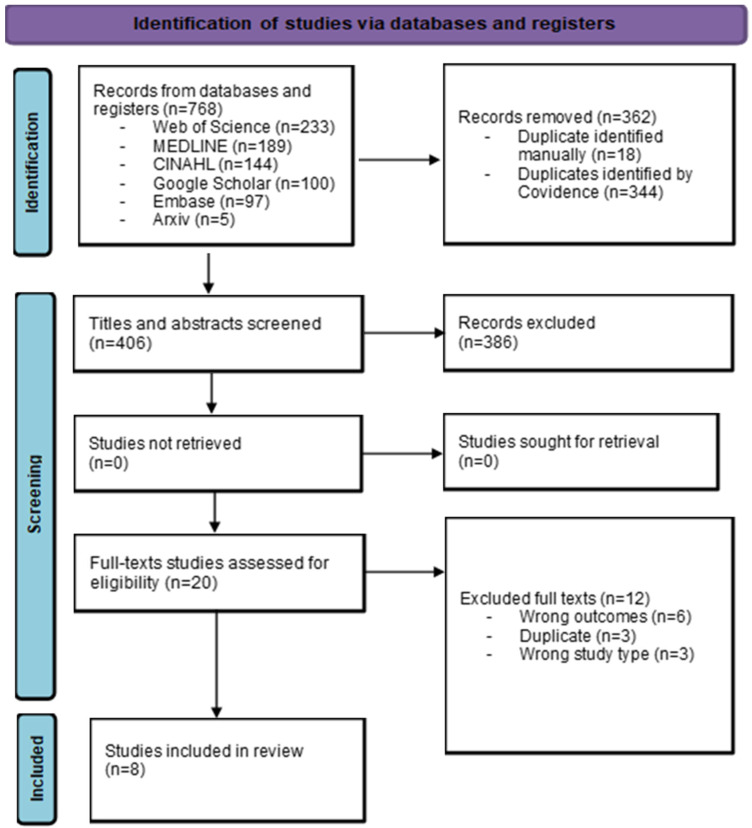
PRISMA 2020 flow diagram.

**Table 1 healthcare-13-01447-t001:** PICO(S) Elements.

PICO(S) Elements [26]	Elements in this Review
Participants	All healthcare providers engaged in clinical documentation
Intervention	AI tools designed to streamline clinical documentation, including: Real-time transcription and notetaking Automated data entry into EHRs Natural Language Processing (NLP) for clinical summarization, contextual understanding, and structured documentation Tools transforming spoken interactions into organized clinical notes
Comparison	Usual administrative practices with no additional AI support.
Outcomes	Clinician outcomes: stress, burnout, administrative time reduction (during and after clinical encounter), patient engagement focus, cognitive load. Efficiency outcomes: healthcare system efficiency metrics (e.g., wait times, patient throughput, costs). Documentation outcomes: accuracy, relevance, reduction in manual note revisions. Patient outcomes: quality of care, satisfaction.
Setting	All clinical settings
Specifications for the studies	Eligible studies: Randomized controlled trials (RCTs), quasi-experimental designs, prospective cohort. Studies, pre-post studies, observational, and mixed-method studies. Excluded studies: Protocols, ongoing studies, qualitative-only studies, reviews, commentaries, and non-AI tools.
Language and publication year	No restrictions.

**Table 2 healthcare-13-01447-t002:** Characteristics of included studies.

Studies	Country	Year	Aim of Study	Study Design	Study Setting and Participants	Outcomes Assessed
Haberle et al. [31]	USA	2024	To assess the impact of an ambient listening and digital scribing solution, Nuance Dragon Ambient eXperience (DAX), on caregiver engagement, time spent on Electronic Health Records (EHR), including after-hours use, productivity, panel size for value-based care providers, documentation timeliness, and Current Procedural Terminology (CPT) submissions.	Peer-matched controlled cohort study	Outpatient clinics within an integrated healthcare system. A total of 99 providers from 12 specialties participated. Seventy-six matched control group providers were included in the analysis.	Primary: provider engagement, productivity, panel size, documentation, and coding timeliness. Secondary: patient safety, likelihood to recommend, and number of patients opting out.
Hudelson et al. [32]	USA	2024	(1) To identify and test virtual scribe solutions, both live and asynchronous, tailored to the healthcare system’s needs; (2) to evaluate and implement these technologies to reduce clinicians’ documentation burden, a major contributor to physician burnout.	Mixed methods pilot study	An integrated academic healthcare system. Sixteen clinicians from diverse specialties.	Primary: clinicians’ documentation burden, clinicians’ overall experience. Secondary: none.
Islam et al. [33]	Bangladesh	2024	(1) To develop an automated scribe and intelligent prescribing system for health professionals by identifying user requirements; (2) to design a system that generates medical notes and prescriptions efficiently from voice commands, enhancing the usability of digital scribe solutions; and (3) to evaluate the system’s performance to ensure it meets clinicians’ needs and improves documentation processes.	AI system development process. Post-test questionnaire	A Medical College Hospital. Enlisted the participation of 17 diabetes patients and six doctors.	Primary: similarity rates between AI scribes and prescriptions compared to those generated manually, the system’s usability. Secondary: none.
Kernberg et al. [34]	USA	2024	To evaluate the accuracy and quality of Subjective, Objective, Assessment, and Plan (SOAP) notes generated by ChatGPT-4 using established History and Physical Examination transcripts as the gold standard, identifying errors and assessing performance across categories.	Comparative Study	Fourteen simulated patient-provider encounters, including professional standardized patients, represented a wide range of ambulatory specialties and two clinical experts.	Primary: an AI model’s (ChatGPT-4) performance evaluation (e.g., variations in errors, accuracy, and quality of notes generated) using established transcripts of “History and Physical Examination” as the gold standard. Secondary: none.
Nguyen et al. [35]	USA	2023	To pilot a digital scribe in live clinic settings at a National Cancer Institute–designated Comprehensive Cancer Center, assess its impact on clinician well-being and documentation burden, and identify facilitators and barriers to effective implementation.	Mixed-methods longitudinal pilot study	Clinic settings at a National Cancer Institute–designated Comprehensive Cancer Center, evaluated by 21 “clinician champions”.	Primary: Impact on clinician well-being and documentation burden, implementation facilitators and barriers for effective AI scribe use, feasibility, and usability. Secondary: Clinicians reported some patients expressed unease at having their visits recorded on a smartphone.
Sezgin et al. [36]	USA	2024	To present a proof-of-concept digital scribe system for summarizing Emergency Department consultation calls to support clinical documentation and report its performance.	Usability Study. Quantitative descriptive.	Nationwide Children’s Hospital Physician Consult and Transfer Center. One hundred phone call recordings from 100 unique callers (physicians) for ED referrals are used.	Primary: Performance (e.g., accuracy rates in medical records, ability to comprehend and replicate the structure and flow of clinical dialogue) of four pre-trained large language models (T5-small39, T5-base39, PEGASUSPubMed47, and BART-Large-CNN46) to support clinical documentation. Secondary: None.
van Buchem et al. [37]	Netherlands	2023	To assess the impact of a Dutch digital scribe system on clinical documentation efficiency and quality.	Usability Study	Leiden University Medical Center. Twenty-two medical students with experience in clinical practice and documentation.	Primary: Clinical documentation efficiency (i.e., summarization time) and quality (e.g., accuracy, usefulness). Secondary: None.
Wang et al. [38]	USA	2021	To develop a digital scribe for automatic medical documentation using patient-centered communication elements.	Simulation of patient encounters. Quantitative descriptive.	Across multiple departments within a university medical center with two medical students.	Primary: efficiency, training required, documentation speed, patient-centered communication, and reliability. Secondary: None.

**Table 3 healthcare-13-01447-t003:** Characteristics of AI scribe interventions.

Studies	Type of AI Scribe (Name)	Key Global Characteristics	How the System Works					
			CD	RT	AN	ADE	AS	AA
Haberle et al. [31]	Mobile App (Nuance DAX)	An AI-powered, voice-enabled solution that automatically documents clinical encounters using ambient listening and conversational AI to generate comprehensive documentation from patient-provider conversations.	X	X	X	X	X	X
Hudelson et al. [32]	Mobile App (Not disclosed)	Two virtual scribe solutions were compared: (1) Live Virtual Scribe (All Human-Driven) and (2) Asynchronous Virtual Scribe (Hybrid AI/Human), which uses audio recordings, machine learning, and NLP to pre-populate notes, then reviewed and finalized by a human scribe within 4 h, working asynchronously.	X	X	X	X		
Islam et al. [33]	Software (Not disclosed)	The system converts patient voice descriptions into text to generate medical notes and uses extracted medical terms for this purpose. It also creates e-prescriptions from doctors’ voice commands. Using NLP and machine learning, it records medical information and generates prescriptions based on voice input from healthcare professionals.	X	X	X	X	X	X
Kernberg et al. [34]	Chatbot (Not disclosed)	The ChatGPT–4–generated Subjective, Objective, Assessment, and Plan (SOAP) format is a standard clinical documentation model that organizes interview data into structured headers. It provides a clear framework for healthcare professionals to record and share patient information.	X		X		X	
Nguyen et al. [35]	Mobile App (Dragon Ambient eXperience)	DS smartphone app: The digital scribe’s AI components structured the recorded information into a visit note, and the vendor’s staff performed initial editing before the notes were released to the clinician.	X		X			
Sezgin et al. [36]	Pre-trained large language models (Not disclosed)	The system converts audio recordings into text: AWS Transcribe transcribes the audio, and an annotator reviews and corrects the transcript. Transcription documents are organized as text input for the model, with nurse summary notes used as reference. Four pre-trained language models (T5-small, T5-base, PEGASUS-PubMed, and BART-Large-CNN) are employed to summarize clinical conversations based on their strengths in the healthcare domain.					X	
van Buchem et al. [37]	Software (Autoscriber)	A web-based tool that transcribes and summarizes medical conversations in Dutch, English, and German. It uses a transformer-based speech-to-text model fine-tuned to clinical data, along with large language models like GPT-3.5 and GPT-4, and a custom prompt structure for summarization. The tool also features self-learning functionality, which was not evaluated in this study.		X			X	
Wang et al. [38]	Mobile App/Software (Not disclosed)	It uses automatic speech recognition and natural language processing to transcribe and summarize conversations between healthcare providers and patients into written text.	X	X	X	X	X	

**Table 4 healthcare-13-01447-t004:** Impact of AI scribes on each outcome category.

Categories of Outcomes	Impacts of AI Scribes
(1) Clinician outcomes (e.g., experience with the tool, stress, burnout, documentation burden, etc.)	Provider engagement: - Positive trends in engagement among AI scribe (DAX) users, with a score of 3.62 vs. 3.37 for non-users, as measured by the Press Ganey Workforce Engagement surveys (5-point Likert scale) [31]. Clinician documentation burden: - Metrics assessed through EHR metadata and surveys show a decreased documentation burden for some clinicians, suggesting AI scribes may be effective [32]. Users’ overall experience: - Feedback from pre- and post-intervention surveys and semi-structured interviews was mixed. Some clinicians provided positive feedback, while others expressed concerns about AI scribe training and quality [32]. - The system was generally found to be user-friendly, intuitive, and easy to use [33]. - Usability, feasibility, and acceptability were assessed using a REDCap survey, with marginal scores for feasibility (16.0), acceptability (16.3), and usability (68.6) [35]. However, the system showed limited improvements in reducing burnout, with no significant change in burnout (*p* = 0.081) but improved perceptions of documentation time (*p* = 0.005). - User feedback: Medical students found the AI scribe (Autoscriber) useful and easy to use but criticized the length and rigid structure of the summaries. During editing, medical students often added context and removed irrelevant text [37].
(2) Healthcare system efficiency (e.g., wait times, patient throughput, costs, etc.)	Productivity (the volume and intensity of clinical services provided by healthcare providers): - Statistically significant increase in work Relative Value Unit (wRVU) productivity for DAX users (94.2% vs. 90.6% at baseline) [31]. Data were gathered from March to July 2022, with historical data included for comparison. The results showed positive trends in provider engagement, whereas non-participants experienced a decline in engagement. Additionally, there was a statistically significant, though modest, increase in productivity, which, however, had limited practical significance. Attributed panel size (the number of patients assigned to a value-based care (VBC) provider): - No significant change in panel size for VBC providers (773 vs. 776 at baseline) [31]. For primary care providers in value-based care (VBC), the average panel size at baseline was 776 for the study cohort and 752 for the control cohort. By the end of the study, these averages were 773 and 742, respectively. A comparison of the baseline and study conclusion data showed no significant difference (*p* = 0.17).
(3) Documentation outcomes (e.g., accuracy, relevance, deficiency rates, etc.)	Documentation time and EHR usage: - Documentation time per patient decreased from 5.3 min to 4.54 min for AI scribe (DAX) users, but after-hours EHR work increased by 4.69% for DAX users, compared to a decrease of 0.945% for the control group [31]. Documentation deficiency rates: - There was a decrease in the 24 h documentation deficiency rate (8.6% to 6.3%), which was statistically significant [31]. - There was an increase in the 24 h CPT submission deficiency rate (27.9% to 30.0%), which is also statistically significant [31]. Similarity rates: - System-generated outputs were evaluated for similarity to manually created notes: 87.5% for scribes and 96.2% for prescriptions, indicating promising results in enhancing documentation efficiency and accuracy [33]. ChatGPT-4 evaluation: - ChatGPT-4 showed substantial variability in errors, accuracy, and note quality, struggling particularly with non-objective data. These inconsistencies suggest that it currently cannot meet clinical standards for documentation [34]. Error and usability insights: - Errors were noted in parts of the note (e.g., patient education), with improvements in the editing workload over time [35]. Documentation delays were reported due to the vendor release process before signing. Pre-trained models’ evaluation: - Summarizations using AWS Transcribe and pre-trained models showed that BART-Large-CNN had an F1 ROUGE-1 score of 0.49, with 71.4% recall and 67.7% accuracy in identifying key information. Performance declined by over 50% when a zero-shot approach was used [36]. Impact of a commercial digital scribe system (Autoscriber): - Time efficiency was improved, with a median time for manual summarization at 202 s and for editing automatic summaries at 186 s. However, automatic summaries had lower Physician Documentation Quality Instrument (PDQI-9) scores and higher word counts compared to manual summaries, suggesting room for improvement [37]. Individual variability: - Impact on documentation quality (PDQI-9) and time spent varied among medical students. Some students spent more time editing automatic summaries than manually creating them [37]. Patient-centered digital scribe: - A developed digital scribe was found to be 2.7 times faster than typing, 2.17 times faster than dictation for history sections, and 3.12 times faster for physical exams [38]. It showed higher efficiency and reliability compared to traditional methods, with minimal training required.
(4) Patient outcomes (e.g., safety or quality of care, experience with technology, etc.)	Patient safety: - No reported patient safety events related to AI scribe (DAX) use and no significant change in Likelihood to Recommend (LTR) scores [31]. Patient experience with technology: - Some patients expressed unease with recording via smartphones [35]. Patient-centered communication: - A developed digital scribe using patient-centered communication techniques showed promise in enhancing patient-provider communication while maintaining effective documentation [38]. These techniques allowed providers to document relevant information without disengaging from the patient.

**Table 5 healthcare-13-01447-t005:** Factors for the Successful Adoption and Implementation of AI Scribes in Clinical Settings.

Categories of Factors	Items for Successful Adoption and Implementation of AI Scribes in Clinical Settings
(1) Training and support needs	- Continuous education and onboarding are essential for clinical users [31]. - Comprehensive training programs should align with institutional priorities and have executive sponsorship [32]. - Individualized training, on-site support, and clinician super-users improve workflow integration [35]. - Variability in performance suggests that a commercial digital scribe system may benefit some users more than others [37]. - A developed patient-centered digital scribe system found that minimal training is required, with users improving their proficiency over time, indicating strong potential for broad clinical adoption [38].
(2) Organizational preparation	- Engaging key clinical stakeholders from the project’s inception is crucial [31]. - Implementation should follow a collaborative, iterative selection process aligned with institutional priorities [32]. - Caution is advised before the widespread adoption of AI-generated medical documentation (e.g., ChatGPT-4) due to variability in model performance and limitations [34]. - Inconsistent model performance highlights the need for user-centered design and systems thinking to address technical challenges [36].
(3) Technical considerations and improvements	- AI scribes should be customizable to meet healthcare providers’ needs, though this requires additional time and resources [31]. - Ensuring interoperability with electronic health records (EHRs) is essential [32,33]. - Features such as real-time transcription, automated notetaking, intuitive user interface, and accurate medical term extraction should be prioritized [33]. - Limitations include ChatGPT-4’s lack of real-time feedback, limiting learning from user interactions [34], variability in error types, and the inability to adjust temperature settings, which impacts output consistency [34]. - External factors like background noise and speaker inconsistencies affect transcription accuracy [36], while discrepancies between audio summaries and intake notes, particularly for follow-up details, need addressing [36]. - AI-generated summaries tend to have a higher word count and lower lexical diversity, potentially affecting downstream clinical processes [37]. - Future iterations could benefit from customizable verbal cues and improvements like reducing vendor turnaround time, standardizing consent processes, and developing web-based interfaces for exam rooms [35,38].
(4) Evaluation and workflow integration	- Many providers were unaware of AI scribes’ capabilities or lacked trust in their accuracy in documenting patient encounters [31]. - Ongoing evaluation and continuous feedback mechanisms are necessary for successful adoption [32]. - Standardized operating procedures and checklists could enhance adoption [35]. - While AI scribes can improve documentation efficiency, they require better workflow integration and error management [35]. - Inconsistent model performance (e.g., BART-Large-CNN) indicates a need for improved recognition capabilities, and documentation variations suggest differences in nurses’ notetaking styles [36].
(5) Ethical considerations	- Concerns exist about the accessibility and transparency of AI scribes, as well as the potential misinterpretation of medical terminology, leading to incorrect documentation [31]. - Some clinicians avoided AI scribes in shared office spaces due to privacy concerns, and patients with speech impairments may be excluded, raising equity concerns [35]. - Evidence-based guidelines are needed to ensure ethical and secure AI use in healthcare, with implementation prioritizing data privacy, patient consent, and seamless integration with electronic medical records [36,38].
(6) Further research and future directions	- Studies should evaluate the impact of more advanced technologies (e.g., DAX Copilot, an advanced version of Nuance DAX) and compare personalization or efficiency training with ambient AI documentation [31]. - Larger-scale studies with control groups are needed to assess AI’s impact on clinician workload and efficiency [32]. - Research is required for real-world implementation, scalability, and improvements in AI accuracy, reducing variability and minimizing errors [33,34]. - Investigating real-time training models and support mechanisms could enhance adoption while expanding research to include hybrid approaches and language models would be beneficial [35,36]. - Comparative studies should assess documentation burden and human factors [36], with a focus on user-specific benefits based on age, specialty, and typing skills [37]. - Future studies should also explore AI’s impact on physician-patient interactions, workflow, quality of care, and satisfaction [37], as well as the integration of patient-centered communication to reduce physician burnout [38]. - Continuous evaluation is necessary as language models evolve rapidly, and further customization may optimize AI scribe communication styles for individual users [37,38].

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
