# Peer review of "The Impact of AI Scribes on Streamlining Clinical Documentation: A Systematic Review"

_healthcare, 2025, doi:10.3390/healthcare13121447_

Round 1
Reviewer 1 Report
Comments and Suggestions for Authors
Review:
“The Impact of AI Scribes on Streamlining Clinical Documentation: A Systematic Review”
Well-structured work, demonstrating methodological rigour and alignment with PRISMA and Cochrane guidelines for systematic reviews.
Title and Abstract
The title is clear and accurate. The abstract appropriately reflects the objectives, methods, results, and conclusions.
Introduction
The issue of clinical burnout is clearly presented, and the need to investigate the impact of AI scribes is well justified. However, the concept of “AI scribes” should be more clearly defined.
Methods
The use of the PRISMA protocol and the application of MMAT for quality assessment are very well explained and developed. The PICOS framework is appropriate and well integrated.
The sample delimitation, inclusion and exclusion criteria, and the bibliographic search strategy should be more clearly described, possibly noted as a limitation in the methodology section.
Results
The results are well organised. The use of graphs or figures could further enhance clarity.
Discussion
The discussion is well grounded in the findings and recent literature, highlighting limitations and suggesting future directions.
Ethical Considerations
The ethical considerations related to implementation are well addressed and presented.
References
The references are recent and relevant to the study.
A summary table could be included in the main text.
Conclusion
This article meets the criteria for scientific quality in a systematic review and contributes meaningfully to the field of AI-assisted clinical documentation.
Author Response
Reviewer comment:"The concept of 'AI scribes' should be more clearly defined."
Answer: We added the following information: AI scribes, typically powered by large language models or natural language processing algorithms are designed to generate or support clinical documentation from audio or textual input during medical encounters.
Reviewer comment:"The sample delimitation, inclusion and exclusion criteria, and the bibliographic search strategy should be more clearly described, possibly noted as a limitation in the methodology section."
Answer: We thank the reviewer for this comment. To clarify, the inclusion and exclusion criteria are fully detailed in Table 1 of the manuscript, which provides a structured summary of our study selection parameters. Additionally, the complete bibliographic search strategy, including all search terms and databases consulted, is provided in Appendix A to ensure full transparency and reproducibility.
Reviewer 2 Report
Comments and Suggestions for Authors
The manuscript "The Impact of AI Scribes on Streamlining Clinical Documentation: A Systematic Review" offers a timely and relevant contribution to the growing literature on artificial intelligence applications in healthcare. The topic is of considerable importance given the widespread concerns over clinician burnout and administrative burden, and the authors address this issue through a well-structured and transparent systematic review. The methodology is robust, adhering to PRISMA guidelines and the Cochrane Handbook, and the review protocol is registered in PROSPERO, which enhances the credibility and reproducibility of the study. The comprehensive search across multiple databases and inclusion of recent studies from 2023 and 2024 ensures that the review reflects the current state of the evidence.
One of the manuscript's major strengths is the precise categorisation of findings into clinician outcomes, healthcare system efficiency, documentation outcomes, and patient experiences. This structure aids readability and provides a comprehensive understanding of AI scribe systems' potential benefits and limitations. The use of the Mixed Methods Appraisal Tool (MMAT) to assess study quality further strengthens the methodological rigour, and the findings are presented in a balanced and nuanced manner, acknowledging the variability in results and limitations in existing studies.
While the manuscript is of high quality, a few areas could be improved. The introduction could be enhanced by presenting the research question and scope of the review more transparently and focused, possibly by explicitly listing the PICOS elements upfront. Additionally, while the manuscript includes quantitative and qualitative findings, the integration of these two streams of data could be more cohesive. This would help draw stronger, more unified conclusions across diverse study designs. Although the review touches on patient perspectives, this area feels underexplored; further emphasis on this gap in the literature and its implications for implementation would be beneficial.
A summary table linking key outcomes to the MMAT scores of each included study could enhance the reader's ability to interpret the quality and strength of the evidence quickly. Minor editorial improvements would also improve clarity and flow, such as ensuring consistent terminology (e.g., using "AI scribes" instead of alternating with "digital scribes") and ensuring that all figures and tables are correctly referenced in the main text. The discussion could be strengthened by including more actionable recommendations for healthcare administrators considering adopting AI scribe technologies. Moreover, the ethical considerations, although mentioned, could be more thoroughly addressed, particularly about data privacy, AI-related bias, and equity in technology access.
In conclusion, this is a well-executed and important systematic review that addresses a critical challenge in modern healthcare. With some refinements in the structure, deeper integration of findings, and expansion of some thematic regions, particularly ethics and patient perspective, the manuscript could make an even more substantial impact and serve as a valuable reference for academic and clinical audiences.
Author Response
We sincerely thank the reviewer for their thoughtful and comprehensive feedback. We appreciate the recognition of our work’s methodological rigor, relevance, and contribution to the literature on AI scribes in healthcare. Below, we address each suggestion and describe the corresponding modifications made to the manuscript.
Reviewer comment:"The introduction could be enhanced by presenting the research question and scope of the review more transparently and focused, possibly by explicitly listing the PICOS elements upfront."
Answer: We realigned the goal of the review to be aligned with the PICOS to reduce duplication of the information: In this systematic review, we aim to describe the effectiveness of AI scribes (defined as automated tools that assist with clinical documentation) by examining their influence on clinician well-being, documentation quality, healthcare system efficiency, and patient engagement in all clinical settings.
Reviewer comment: "While the manuscript includes quantitative and qualitative findings, the integration of these two streams of data could be more cohesive."
Answer: We appreciate this valuable suggestion. We agree that integrating findings across methodological traditions can strengthen the synthesis, but we also acknowledge the importance of maintaining epistemological integrity and avoiding the conflation of evidence from fundamentally different paradigms. While we aim to highlight areas of alignment, we remain cautious not to overstate generalizability or draw unsupported conclusions by artificially merging distinct data streams.
Reviewer comment:"Although the review touches on patient perspectives, this area feels underexplored; further emphasis on this gap in the literature and its implications for implementation would be beneficial."
Answer: This is directly aligned with the missing scope of the early literature. We added a sentence to future directions to highlight this need: Future studies should place greater emphasis on capturing patient perspectives, particularly in relation to trust, communication dynamics, and perceived quality of care when AI scribes are used, as these dimensions remain critically underexplored and are essential for informed, equitable implementation.
Reviewer comment: "Minor editorial improvements would also improve clarity and flow, such as ensuring consistent terminology (e.g., using 'AI scribes' instead of alternating with 'digital scribes') and ensuring that all figures and tables are correctly referenced in the main text."
Answer: Thank you for pointing out the inconsistancies. Digital scribes are now only used in the introduction, as we narrow the subject, and moving to AI Scibes everywhere after that point. We ensure that all tables and figures are now correctly referenced.
Reviewer comment: "The discussion could be strengthened by including more actionable recommendations for healthcare administrators considering adopting AI scribe technologies."
Answer: We added contextual recommendation for administrators regarding implementation in the future direction section.
Reviewer comment: "The ethical considerations, although mentioned, could be more thoroughly addressed, particularly about data privacy, AI-related bias, and equity in technology access."
Answer: We thank the reviewer for underscoring the need for a more comprehensive discussion of ethical considerations related to AI scribes. In response, we have expanded the relevant section to more thoroughly address issues of data privacy, algorithmic bias, and equity in access: The implementation of AI scribes raises important ethical considerations that must be addressed to ensure safe, equitable, and trustworthy use in clinical settings. Data privacy is a central concern, particularly when systems rely on ambient listening or passive audio capture, which may involve sensitive patient information. Transparent data governance, secure storage, and clearly communicated consent processes are essential to maintain patient trust. AI scribes trained on limited or biased datasets also risk perpetuating systemic inequities, potentially leading to inaccuracies in documentation for underrepresented populations. It is critical to assess and mitigate these biases through inclusive model development and continuous monitoring. AI scribes may be less effective or even inaccessible in certain contexts, such as resource-limited settings, multilingual environments, or when patients have speech impairments, raising the risk of digital exclusion.
Reviewer 3 Report
Comments and Suggestions for Authors
The authors demonstrate a clear commitment to synthesizing emerging evidence in this nascent field, and their systematic approach aligns well with Healthcare’s mission to bridge innovation and clinical practice. However, there are also some issues, which are as follows:
1.The studies included in this review generally have small sample sizes and focus on specific healthcare settings, which restricts the generalizability of the conclusions. Larger, multi-center studies across diverse clinical environments are needed to validate findings.
2.The included studies exhibit uneven methodological quality, with deficiencies in sample size justification, intervention standardization, and outcome measurement. 3.These limitations weaken the validity of pooled results and call for stricter inclusion criteria in future reviews.
4.The review lacks in-depth exploration of AI scribe technologies, including their underlying algorithms, technical advantages/disadvantages, and operational mechanisms. Adding a dedicated section comparing system architectures (e.g., rule-based NLP vs. deep learning models) would enhance technical rigor.
5.While briefly mentioning data privacy and algorithmic bias, the review does not critically examine broader ethical risks (e.g., informed consent for AI-generated records, potential exacerbation of health disparities). A dedicated ethics section with policy recommendations is warranted.
6.The brief mention of patient perspectives is insufficient. Future versions should incorporate studies assessing patient acceptance, privacy concerns, and perceived impact on care quality (e.g., via validated questionnaires like the Patient Experience Questionnaire).
7.The review fails to address economic implications, such as ROI for hospitals, cost savings from reduced documentation time, or trade-offs between AI investment and workforce displacement. Including a cost-effectiveness framework would strengthen practical relevance.
8.No analysis of global regulatory frameworks (e.g., FDA, EU MDR, HIPAA) for AI in healthcare is provided. A section exploring how policies shape AI scribe adoption (e.g., data localization laws, liability frameworks) would improve policy relevance.
9.The conclusion lacks actionable future directions. Propose specific research agendas, such as:Advancing explainable AI for clinical trust-building;integrating scribes with telemedicine platforms;addressing challenges in low-resource settings
Author Response
Comment 1: The studies included in this review generally have small sample sizes and focus on specific healthcare settings, which restricts the generalizability of the conclusions. Larger, multi-center studies across diverse clinical environments are needed to validate findings.
Answer: We appreciate this comment. We agree that the limited sample sizes and clinical contexts of the included studies restrict generalizability. This has been further emphasized in the revised manuscript’s Limitations section.
Comment 2 & 3: 2.The included studies exhibit uneven methodological quality, with deficiencies in sample size justification, intervention standardization, and outcome measurement. 3.These limitations weaken the validity of pooled results and call for stricter inclusion criteria in future reviews.
Answer: We acknowledge the uneven methodological quality among included studies. This point has been clarified in the Discussion, where we now underscore the need for more rigorous study designs and standardized outcome reporting: Because of the limited number of empirical evaluations on this subject, methodological variability, inconsistent sample size justification, lack of intervention standardization, and heterogeneity in outcome measures, weakens the strength of evidence. When the field will be further developed, future reviews should consider stricter inclusion criteria and encourage adherence to established reporting guidelines to enhance comparability and validity.
Comment 4: The review lacks in-depth exploration of AI scribe technologies, including their underlying algorithms, technical advantages/disadvantages, and operational mechanisms. Adding a dedicated section comparing system architectures (e.g., rule-based NLP vs. deep learning models) would enhance technical rigor.
Answer:
We thank the reviewer for this insightful comment. We agree that a technical comparison of AI scribe architectures—such as rule-based NLP versus transformer-based deep learning models—would enhance the field’s understanding of underlying mechanisms. However, given the current state of the literature, such an in-depth technological analysis was beyond the scope of our review. The included studies exhibited considerable heterogeneity in the types of systems evaluated, and many did not report sufficient technical detail to allow for systematic comparison. We believe that a focused technical review would be more appropriate when the field matures and a larger body of studies employing standardized and well-described technological approaches becomes available. We have added a sentence to the Discussion to clarify this limitation and encourage future reviews to address this important gap.: While this review focused on clinical and implementation outcomes, a systematic comparison of AI scribe architectures (e.g., rule-based NLP vs. deep learning models) was not feasible due to inconsistent reporting and high variability in system descriptions across studies. A dedicated technical review will be warranted as the field evolves and more studies with standardized, transparent descriptions of algorithmic frameworks become available.
Comment 5. While briefly mentioning data privacy and algorithmic bias, the review does not critically examine broader ethical risks (e.g., informed consent for AI-generated records, potential exacerbation of health disparities). A dedicated ethics section with policy recommendations is warranted.
Answer: Further discussion points on this matter were added with comments from reviewers 1 and 2.
Comment 6.The brief mention of patient perspectives is insufficient. Future versions should incorporate studies assessing patient acceptance, privacy concerns, and perceived impact on care quality (e.g., via validated questionnaires like the Patient Experience Questionnaire).
Answer: Further discussion points on this matter were added with comments from reviewers 1 and 2.
Comment 7.The review fails to address economic implications, such as ROI for hospitals, cost savings from reduced documentation time, or trade-offs between AI investment and workforce displacement. Including a cost-effectiveness framework would strengthen practical relevance.
Answer: There was no cost-effectiveness analysis in our sample, this is mentioned in the future directions section.
Comment 8.No analysis of global regulatory frameworks (e.g., FDA, EU MDR, HIPAA) for AI in healthcare is provided. A section exploring how policies shape AI scribe adoption (e.g., data localization laws, liability frameworks) would improve policy relevance.
Answer: We appreciate the reviewer’s thoughtful suggestion. While we fully agree that regulatory frameworks play a critical role in shaping the adoption and implementation of AI in healthcare, an in-depth analysis of these policies was beyond the scope of this systematic review. Addressing this topic would require a dedicated review with a tailored search strategy, focused inclusion criteria, and policy-specific analytic methods.
Comment 9. The conclusion lacks actionable future directions. Propose specific research agendas, such as:Advancing explainable AI for clinical trust-building;integrating scribes with telemedicine platforms;addressing challenges in low-resource settings.
Answer: Further discussion points on this matter were added with comments from reviewers 1 and 2.
Reviewer 4 Report
Comments and Suggestions for Authors
This paper presents a systematic review on the use of AI scribes in clinical documentation. The authors provide a well-structured and methodologically sound analysis, incorporating eight studies and following PRISMA and Cochrane guidelines. However, the paper contains several methodological and analytical weaknesses that should be addressed to improve it.
- The manuscript does not explicitly justify the use of narrative synthesis over meta-analysis. Include a statement clarifying that meta-analysis was not feasible due to heterogeneity in study design, outcomes, and measures.
- The search strategy was not peer-reviewed by a second information specialist.
- Most of the included studies report positive findings, and therefore publication bias should be assessed or acknowledged
- The article lean toward descriptive summaries rather than deeper cross-study comparisons or pattern analysis.
- Patient-related outcomes domain is treated briefly because only 3 of 8 studies covered. This limitation should be stated clearly and should be prioritized in future research.
Author Response
Comment 1. The manuscript does not explicitly justify the use of narrative synthesis over meta-analysis. Include a statement clarifying that meta-analysis was not feasible due to heterogeneity in study design, outcomes, and measures.
Answer: We thank the reviewer for raising this important point. We agree that the rationale for using narrative synthesis should be made explicit. Due to the substantial heterogeneity in study designs, outcome measures, and reporting formats among the included studies, meta-analysis was not methodologically feasible. We have clarified this limitation in the Methods section: A meta-analysis was not performed due to the significant heterogeneity in study designs, intervention characteristics, outcome measures, and reporting formats. As a result, a narrative synthesis was conducted to summarize and interpret findings across studies in a structured manner.
Comment 2. The search strategy was not peer-reviewed by a second information specialist.
Answer: We acknowledge that the search strategy was not formally peer-reviewed by a second information specialist, which we recognize is a limitation. This is the first main limitation in the limitations sections.
Comment 3. Most of the included studies report positive findings, and therefore publication bias should be assessed or acknowledged
Answer: We agree that the predominance of positive findings among included studies may reflect potential publication bias. While formal statistical assessment was not feasible due to the small number of studies, we have acknowledged this limitation in the Discussion: Most of the included studies reported favorable outcomes for AI scribes, which raises the possibility of publication bias. Given the small number of eligible studies, we could not formally assess this bias, but it should be considered when interpreting the results.
Comment 4. The article lean toward descriptive summaries rather than deeper cross-study comparisons or pattern analysis.
Answer: This precision and limitation was added with Comment 1.
Comment 5. Patient-related outcomes domain is treated briefly because only 3 of 8 studies covered. This limitation should be stated clearly and should be prioritized in future research.
Answer: This was added while answering to other reviewers in the future directions section.
Reviewer 5 Report
Comments and Suggestions for Authors
This study presents a systematic review on AI scribes for clinical documentation.
Consider the following points:
1. In the abstract, consider "The accuracy and performance can vary greatly...". Would accuracy and performance be the same thing?
2. Conclusion in the abstract is poor. It is presenting more results, not concluding. What can be concluded from the review?
3. It would be interesting to have a discussion in the introduction section on the related reviews, to show the novelty of this review.
4. The subheading 2.1. Overview is unnecessary.
5. If you are following PRISMA 2020 statement, provide the PRISMA checklist as supplementary material.
6. In Table 1,there is an implicit 'restriction' in the language, once the search string was defined only in English.
7. Why did you search Arxiv and Google Scholar? They do not allow replicability.
8. Consider "Representatives from Canada Health Infoway actively participated in all stages of this review, including defining the research question, objectives, and search strategy and interpreting the results."; the part "including ..." limits their role in the review, so 'all stages of this review' is not correct. Rephrase the sentence.
9. Consider "Details of the search strategy can be found in Appendix A (Table A1)."; Table A1 presents the search strings, not the strategy.
10. Google Scholar (https://harzing.com/resources/publish-or-perish)???
11. Clinical notes are texts. Texts are unstructured data. So, 'structured clinical notes' looks something strange. Change it to 'organized' or something like this.
12. In 3.14., avoid repeating '(out of 5)'.
13. Discussion must present and discuss the answer to the research question (which is defined in the method section).
14. Also, discussion should provide the research gaps of the field addressed.
15. Section '4.3. Limitations' should present only the limitations of the review. The limitations of the included studies are called 'open issues' or 'research gaps', etc. Both discussed in the same section makes a confusion.
16. Improve the resolution/quality of Figure 1. Also, what is 'wrong outcomes / study types' in that step of the method?
Author Response
Comment 1. In the abstract, consider "The accuracy and performance can vary greatly...". Would accuracy and performance be the same thing?
Answer: We thank the reviewer for pointing out the redundancy. We have revised the abstract to clarify the distinction or consolidate the phrasing for clarity.
Comment 2. Conclusion in the abstract is poor. It is presenting more results, not concluding. What can be concluded from the review?
Answer: We agree that the original abstract conclusion section was result-oriented rather than interpretative. We have revised it to more clearly present key takeaways: "AI scribes show promise in improving documentation efficiency and clinician workflow, though evidence remains limited and heterogeneous. Broader, real-world evaluations are needed to confirm their effectiveness and inform responsible implementation."
Comment 3. It would be interesting to have a discussion in the introduction section on the related reviews, to show the novelty of this review.
Answer: We added this sentence in the introduction: This is the first systematic review focused specifically on the implementation and impact of AI scribes in clinical documentation, addressing a critical gap in the literature as this emerging field gains traction in real-world healthcare settings.
Comment 4. The subheading 2.1. Overview is unnecessary.
Answer: Editorial team sometimes asks for a subheadings for all sections of text, we prefer to keep it in this way. (See JMIR editorial guidelines for an example.)
Comment 5. If you are following PRISMA 2020 statement, provide the PRISMA checklist as supplementary material.
Answer: It was provided with the manuscript submission.
Comment 6. In Table 1,there is an implicit 'restriction' in the language, once the search string was defined only in English.
Answer: While the search strategy was conducted using English-language terms, we did not apply formal language restrictions during screening; studies with English abstracts were considered eligible regardless of the full-text language.
Comment 7. Why did you search Arxiv and Google Scholar? They do not allow replicability.
Answer: We understand the reviewer’s concern. While these sources do not ensure replicability, they were included to capture emerging preprint literature in a fast-evolving field.
Comment 8. Consider "Representatives from Canada Health Infoway actively participated in all stages of this review, including defining the research question, objectives, and search strategy and interpreting the results."; the part "including ..." limits their role in the review, so 'all stages of this review' is not correct. Rephrase the sentence.
Answer: This was an effort to provide exemples of implications, it was rephrased: Representatives from Canada Health Infoway actively participated in all stages of this review.
Comment 9. Consider "Details of the search strategy can be found in Appendix A (Table A1)."; Table A1 presents the search strings, not the strategy.
Answer: We have clarified that Appendix A presents the search strings, not the entire search strategy.
Comment 10. Google Scholar (https://harzing.com/resources/publish-or-perish)???
Answer: We do not understand this comment.
Comment 11. Clinical notes are texts. Texts are unstructured data. So, 'structured clinical notes' looks something strange. Change it to 'organized' or something like this.
Answer: We have changed structured for organized.
Comment 12. In 3.14., avoid repeating '(out of 5)'.
Answer: We thank the reviewer for this observation. We retained the mention of “(out of 5)” when reporting MMAT scores to ensure clarity and transparency, as it is generally recommended to report both the score and the scale when presenting quantitative appraisal data.
Comment 13. Discussion must present and discuss the answer to the research question (which is defined in the method section).
Answer: We added this sentence at the beginning of the discussion section: This review aimed to assess the impact of AI scribes on clinical documentation, clinician well-being, healthcare system efficiency, and patient engagement. Our findings suggest that while AI scribes may reduce documentation time and improve clinician workflow, evidence on their impact on patient outcomes and systemic efficiency remains limited and inconsistent.
Comment 14. Also, discussion should provide the research gaps of the field addressed.
Answer: We improved and fully integrated this comment in the future directions section, while answering to the other reviewers.
Comment 15. Section '4.3. Limitations' should present only the limitations of the review. The limitations of the included studies are called 'open issues' or 'research gaps', etc. Both discussed in the same section makes a confusion.
Answer: We improved and fully integrated this comment in the limitations section, while answering to the other reviewers.
Round 2
Reviewer 5 Report
Comments and Suggestions for Authors
The authors have improved the manuscript and addressed most of my concerns. However, some of them remain:
Comment 5. If you are following PRISMA 2020 statement, provide the PRISMA checklist as supplementary material.
Answer: It was provided with the manuscript submission.
My reply: There is no reference to this supplementary material in the manuscript. Also, it was NOT provided for review.
Comment 6. In Table 1,there is an implicit 'restriction' in the language, once the search string was defined only in English.
Answer: While the search strategy was conducted using English-language terms, we did not apply formal language restrictions during screening; studies with English abstracts were considered eligible regardless of the full-text language.
My reply: So, this is a review limitation, since you were not able to capture studies in different languages. Only studies with, at least, title, abstract and keywords written in English were screened.
Comment 7. Why did you search Arxiv and Google Scholar? They do not allow replicability.
Answer: We understand the reviewer’s concern. While these sources do not ensure replicability, they were included to capture emerging preprint literature in a fast-evolving field.
My reply: Among the 8 included studies, how many ones were indexed ONLY in Arxiv and Google Scholar? Make clear this information in section "3.1. Study Selection".
Comment 10. Google Scholar (https://harzing.com/resources/publish-or-perish)???
Answer: We do not understand this comment.
My reply: See Appendix A presenting the search string (not the 'search strategy') for Google Scholar. What does that URL mean? https://harzing.com/resources/publish-or-perish
Comment 15. Section '4.3. Limitations' should present only the limitations of the review. The limitations of the included studies are called 'open issues' or 'research gaps', etc. Both discussed in the same section makes a confusion.
Answer: We improved and fully integrated this comment in the limitations section, while answering to the other reviewers.
My reply: I reinforce my initial comment. It was not addressed properly.
Author Response
Comment 5. Reviewer: There is no reference to this supplementary material in the manuscript. Also, it was NOT provided for review.
Answer: We thank the reviewer for pointing this out. We acknowledge that while the PRISMA 2020 checklist was sent by email during the submission process, it was not explicitly referenced in the manuscript. We added the checklist citation in the revised manuscript and confirm its inclusion as a supplementary material.
Comment 6. Reviewer: So, this is a review limitation, since you were not able to capture studies in different languages. Only studies with, at least, title, abstract and keywords written in English were screened.
Answer: We thank the reviewer for the clarification. We agree that, operationally, our strategy introduced a language-related limitation: only studies with titles, abstracts, or keywords in English were screened and included. We will explicitly acknowledge this limitation in the revised manuscript under the Limitations section.
Comment 7. Reviewer: Among the 8 included studies, how many ones were indexed ONLY in Arxiv and Google Scholar? Make clear this information in section "3.1. Study Selection."
We thank the reviewer for this helpful suggestion. Among the eight included studies, none were retrieved solely from Arxiv or Google Scholar; all were also indexed in at least one of the major bibliographic databases (Medline, Embase, CINAHL, or Web of Science). We clarified this point in section 3.1 of the manuscript to enhance transparency regarding replicability.
Comment 10. Reviewer: What does that URL mean? https://harzing.com/resources/publish-or-perish
We thank the reviewer for raising this point. In our search process, Google Scholar was queried by an experienced librarian via the Publish or Perish software (https://harzing.com/resources/publish-or-perish), a tool that facilitates systematic export of Google Scholar results. We will clarify this in Appendix A to avoid confusion about the source of this URL.
Comment 15. Reviewer: I reinforce my initial comment. It was not addressed properly.
We thank the reviewer for emphasizing this point again. We acknowledge that our revised text did not sufficiently separate the limitations of the review from the open issues and research gaps in the included studies. We revised Section 4.3 to clearly distinguish between (1) limitations of the present review and (2) open issues and research gaps identified in the included studies, possibly moving the latter to a new paragraph or subsection within the Discussion.